# Advancements in Photothermal Therapy Using Near-Infrared Light for Bone Tumors

**DOI:** 10.3390/ijms25084139

**Published:** 2024-04-09

**Authors:** Mengzhang Xie, Taojun Gong, Yitian Wang, Zhuangzhuang Li, Minxun Lu, Yi Luo, Li Min, Chongqi Tu, Xingdong Zhang, Qin Zeng, Yong Zhou

**Affiliations:** 1Department of Orthopedic Surgery and Orthopedic Research Institute, West China Hospital, Sichuan University, Chengdu 610041, China; xiemengzhang23@163.com (M.X.); taojungong@163.com (T.G.); wangytbone1991@163.com (Y.W.); lizzhuang@163.com (Z.L.); minhun@126.com (M.L.); huaxiluoyi@163.com (Y.L.); minli1204@scu.edu.cn (L.M.); tucq@scu.edu.cn (C.T.); 2National Engineering Biomaterials, Sichuan University Research Center for Chengdu, Chengdu 610064, China; zhangxd@scu.edu.cn; 3NMPA Key Laboratory for Quality Research and Control of Tissue Regenerative Biomaterials, Institute of Regulatory Science for Medical Devices, National Engineering Research Center for Biomaterials, Sichuan University, Chengdu 610064, China

**Keywords:** phototherapy, photothermal therapy, bone cancer, tumor therapy, nanoparticles

## Abstract

Bone tumors, particularly osteosarcoma, are prevalent among children and adolescents. This ailment has emerged as the second most frequent cause of cancer-related mortality in adolescents. Conventional treatment methods comprise extensive surgical resection, radiotherapy, and chemotherapy. Consequently, the management of bone tumors and bone regeneration poses significant clinical challenges. Photothermal tumor therapy has attracted considerable attention owing to its minimal invasiveness and high selectivity. However, key challenges have limited its widespread clinical use. Enhancing the tumor specificity of photosensitizers through targeting or localized activation holds potential for better outcomes with fewer adverse effects. Combinations with chemotherapies or immunotherapies also present avenues for improvement. In this review, we provide an overview of the most recent strategies aimed at overcoming the limitations of photothermal therapy (PTT), along with current research directions in the context of bone tumors, including (1) target strategies, (2) photothermal therapy combined with multiple therapies (immunotherapies, chemotherapies, and chemodynamic therapies, magnetic, and photodynamic therapies), and (3) bifunctional scaffolds for photothermal therapy and bone regeneration. We delve into the pros and cons of these combination methods and explore current research focal points. Lastly, we address the challenges and prospects of photothermal combination therapy.

## 1. Introduction

Bone cancer typically encompasses malignant tumors of the bone, which are categorized as either primary or metastatic bone tumors based on their tumor origin. Among these, primary bone tumors, notably osteosarcoma, are prevalent among adolescents and young adults 10 to 30 years of age, with a peak incidence [1,2]. The indications for limb salvage surgery have expanded and it has become the predominant surgical approach due to advancements in medical technology and biomedical engineering including the development of 3D printing technology and implant materials [3,4]. En bloc tumor resection and bone defect reconstruction are utilized for primary malignant tumors, while metastatic bone tumors are treated with tumor palliative resection and bone defect management [5], significantly impacting the patients’ quality of life. However, regardless of the surgical approach employed, tumor recurrence or residual presence [4] are still clinical challenges due to residual tumor cells still existing and forming in a dormant state after en bloc tumor resection as the seeds of relapse. Additionally, the efficacy of chemotherapy and other comprehensive treatments is suboptimal, resulting in the survival of cells that have spread beyond the tumor capsule. The survival rate among patients experiencing postoperative tumor recurrence is below 20%, thereby exposing them to the potential for a subsequent amputation [6]. This outcome significantly diminishes the patients’ quality of life, amplifies societal health economic costs, and imposes a substantial burden on families.

To mitigate the reoccurrence of bone tumors, conventional treatments such as chemotherapy and radiotherapy are commonly employed, alongside emerging approaches like ablation [7] (microwave, cryotherapy, radiofrequency, etc.), targeted therapy, and immunosuppression. Nevertheless, certain bone tumors exhibit resistance to radiotherapy; patients have local or systemic adverse reactions subsequent to radiotherapy, chemotherapy, and drug administration. Ablation is primarily applied in benign tumors, whereas malignant bone tumors cause elevated postoperative complications, thereby inflicting harm upon the adjacent healthy tissues [8]. It is therefore imperative to advance the development of novel clinical therapies that possess non-invasive attributes, high efficiency, and a commendable safety profile, with the aim of mitigating recurrence rates and fostering the process of osteogenesis. It is necessary to explore strategies for achieving an efficient and secure approach to eradicating bone tumor cells subsequent to surgical intervention for bone tumors.

In recent years, the photothermal therapy (PTT) of tumors has attracted considerable attention due to its minimal invasiveness and high selectivity [1,9,10]. Phototherapy (PT), as its name implies, involves the utilization of light for therapeutic purposes in the treatment of various diseases. Centuries ago, ancient civilizations employed sunlight as a remedy for skin ailments such as psoriasis, and leukodystrophy. In contemporary times, the advent of laser technology has enabled individuals to exploit its localized intense heat for cellular ablation, albeit with significant constraints in tumor treatment. These limitations primarily stem from the laser’s remarkable specificity in targeting malignant cells and the necessity for a light source with a high power density to induce cellular death. Through advancements in contemporary medical technology and nanometer materials, it has been discovered that the utilization of light-based modalities holds potential for tumor treatment [11].

Phototherapy (PT), which includes photodynamic therapy (PDT) and PTT, relies on the efficient conversion of light energy into heat energy by the photothermal agent (PTA) to exert cytocidal effects on tumor cells (Figure 1). This is due to the heightened sensitivity of cancer cells, compared to normal cells, to elevated temperatures. When the local temperature reaches or exceeds 42 °C, certain thermoresistant cellular proteins (antishock proteins) undergo denaturation, resulting in physical alterations to chromatin and the inhibition of DNA synthesis and repair [12]. Consequently, cancer cells ultimately perish while normal tissues endure. This antitumor method is a minimally invasive and highly effective.

PTAs include organic dye molecules, organic nanoparticles, noble metal materials, carbon-based materials, and other inorganic materials [11] (Figure 2). PTAs are important factors in PTT, and can effectively absorb and convert light energy into heat under a light source with a specific wavelength. They absorb energy from photons and transform from a ground state singlet to an excited singlet state. The return to the ground state is modulated by collisions between the excited PTA and its surrounding molecules. This increase in kinetic energy results in the heating of the surrounding microenvironment.

PDT is a therapeutic approach that involves the utilization of a photosensitizer (PS) and optical activation techniques with minimal invasiveness for the treatment of tumor-related ailments. A PS, when selectively accumulated in tumor tissues, can be activated by specific nonthermal wavelengths of light to generate reactive oxygen species (ROS) (Figure 3), particularly singlet oxygen [17]. This process has direct cytotoxic effects on cancer cells, leading to apoptosis, necrosis, or autophagy, ultimately resulting in cytotoxicity and cell death [18]. PDT requires three components: a photosensitizer, molecular oxygen, and light [11]. Although PDT has been applied in treating superficial tumors such as skin cancer and esophageal cancer, its application faces several limitations in the context of bone tumors. These include the following: (1) limited light penetration: most PDT light sources have a shallow penetration depth, which poses challenges for treating deep-seated bone tumors [19,20]; (2) selective accumulation challenges: PSs used in PDT are primarily organic small molecules, and there is a need for further investigation into their selective accumulation in tumor cells [9,20]; (3) the therapeutic effects are oxygen dependent, but most bone tumors are characterized by a hypoxic microenvironment [11]. This issue contributes to the low production of reactive oxygen species (ROS), and consequently, a weakened toxic antitumor effect. For PTT, although they are not used clinically, many PTAs and their corresponding light sources have been explored. When the near-infrared wavelength is in the window of biological tissue, it can penetrate biological soft tissue to a depth that enables noninvasive tumor treatment [21,22]. Moreover, as the scientific understanding of nanomaterials [23] has continued to deepen, it has been shown that nanotechnology can lead to multimodal synergistic therapies by combining various therapeutic elements into a nanoplatform, so it has been widely studied over the past 30 years. In recent years, PTT has been commonly used in combination with other therapies to improve the overall therapeutic effect of bone tumors.

Although most studies are in the preclinical stage, it is still necessary to summarize the related research progress in recent years. In this review, we focus on the latest research on photothermal therapy, classify and summarize studies, and discuss the challenges and opportunities for the application of PTT in the treatment of bone tumors.

## 2. Mechanisms and Characteristics of PTT

### 2.1. Mechanisms of PTT

The biological effects of PTT on cancerous and noncancerous cells are complex and multifaceted, influenced by various factors such as the NIR parameters, the tumor microenvironment, and host immune responses. The biological effects on cells following PTT can include cell necrosis or apoptosis, angiogenesis modulation, inflammatory response, and immune modulation.

The issues with respect to cell necrosis or apoptosis include the following. (1) Direct high-temperature damage: The elevated temperature of tumor cells destroys the cell membrane structure and affects active transport on the cell membrane, disrupting the balance between intracellular and extracellular flow, ultimately leading to cell rupture and death [25]. (2) Intrinsic pathway to cell apoptosis: Cell stress induced by PTT such as DNA damage and heat shock activates the intrinsic pathway, forming apoptosome complexes, triggering caspase-9 activation, and initiating a series of downstream events, ultimately leading to cell apoptosis [26]. (3) A novel perspective is necrosis dangling, suggesting that cancer cell necrosis and apoptosis are temperature-dependent. At temperatures of 41–42 °C, cancer cells release heat shock proteins to mitigate heat damage. However, as the temperature rises to 45 °C, cancer cells undergo necrosis or apoptosis, with cancer cell necrosis occurring at 49 °C [27]. After cancer cell necrosis, the membrane is damaged, releasing intracellular substances into the extracellular matrix. The tumor microenvironment also releases cytokines including calreticulin (CRT), high-mobility group box 1 (HMGB1), adenosine triphosphate (ATP), and tumor-associated antigens, inducing immunogenic cell death (ICD) [28]. Noncancerous cells exhibit lower sensitivity to heat and are less likely to enter a stressed state. However, although noncancerous cells are more tolerant to higher temperatures than cancer cells, they are still affected to some extent. Therefore, precise local heating is a key issue in PTT research, where upconversion nanomaterials have demonstrated promising potential [29].

With respect to angiogenesis modulation, the vascular structure of tumor cells is often disordered, with a low heat response and high blood flow. When the temperature rises under PTT, the endothelial cells of the blood vessel wall are damaged, leading to reduced blood flow. When the temperature reaches 45 °C, tumor cells experience blood stasis and decreased oxygen supply, which can induce cell apoptosis. Moreover, the hypoxic state and reduced blood flow of tumor cells can inhibit their growth and metastasis [30,31]. Conversely, the vascular tissue of noncancerous cells expands under high-temperature conditions, increasing blood flow and promoting heat dissipation, with minimal impact.

### 2.2. Characteristics of PTT

PTT is a method that utilizes the heat effect generated by light-sensitive materials absorbing light energy to treat diseases through the mechanisms above-mentioned. The key lies in the selection of light-sensitive materials; only after passing through these materials can light energy of a certain wavelength be converted into heat energy. The goal of researching this noninvasive therapy is to translate it into clinical applications, thereby alleviating the suffering of tumor patients and improving their survival rates and quality of life. Therefore, it is necessary to focus on the theoretical characteristics that light-sensitive materials should possess when applied in the human body, mainly concerning treatment effectiveness and biological safety. Ideal PTAs possess the following characteristics: (1) excitation wavelength in the NIR range between biological tissue windows I and II; (2) high selectivity for tumor tissues; (3) low toxicity to other tissues and cells, easily soluble in human tissues; (4) relative stability at room temperature; (5) affordable price, simple synthesis, and easy accessibility.

Regarding treatment efficacy, the primary consideration is the absorption in the NIR region of the PTA. This is because NIR wavelengths vary in their penetration depth within the human body. For deep-seated solid tumors, the penetration of NIR to the tumor site to activate the PTA and elicit its effects is a crucial step. Currently, the NIR penetration depth in the human body exceeds that of visible light. The penetration of NIR primarily depends on wavelength and frequency; longer wavelengths penetrate deeper, and higher-frequency NIR also penetrates deeply. However, higher-frequency NIR also increases the risk of laser damage to the surrounding healthy tissues [32]. Currently, the NIR wavelength range mainly includes NIR-I, the first biological window from 650 to 950 nm; NIR-II, the second biological window from 1000 to 1700 nm; NIR-III, from 1600 to 1870 nm; and NIR-IV, from 2100 to 2300 nm. Most PTAs are currently concentrated in NIR-I, while NIR-II offers deeper penetration and has been extensively researched in biological imaging, disease diagnosis, and other areas [33,34]. However, the penetration depth varies among different tissues, necessitating the selection of the most suitable NIR wavelength and corresponding PTA based on the tumor’s location.

Secondly, high photothermal conversion efficiency is crucial as it efficiently enables the conversion of absorbed light energy into heat to elevate the temperature of the targeted tumor tissue. This can also reduce the required dosage and mitigate associated adverse reactions. Biological safety is also paramount. The primary course of a PTA involves its introduction into the human body, reaching the tumor site to exert PTT effects, and subsequent metabolism, absorption, or excretion in the body. Considering these aspects, particle size is a primary consideration for PTAs. Currently, most drugs are administered intravenously, accumulating in tumor cells through enhanced permeability and retention or targeted delivery. However, prior to reaching their destination, they must evade phagocytosis by macrophages and other cells. Additionally, they should be filtered by the liver and kidneys, ensuring sufficient accumulation at the tumor site. Within the human body, PTAs should remain stable under physiological conditions to sustain their efficacy throughout the treatment period. Furthermore, they should exhibit minimal toxicity and possess excellent biocompatibility.

## 3. Nanocarrier Particles—Target Antitumor Treatment with PTT

The success of PTT in treating bone tumors hinges on the accumulation concentration of PTAs within the tumors. Systemic administration of PTAs results in varied accumulation properties within tumor cells due to the diverse structures and physical-chemical characteristics of PTAs. To optimize photothermal treatment outcomes, it has been proposed that nanomaterials can be leveraged as carriers.

Recent advancements in the study of nanomaterials [35], encompassing their structure and size properties, have unveiled several advantages. These include their nanoscale size, promising drug release characteristics, high surface-to-volume ratio, and inherent antitumor cell effects, rendering them suitable carriers. Through surface modifications, these carriers can precisely target tumor cells, ensuring efficient delivery and enhancing distribution and metabolism in vivo [36,37,38]. Notably, targeted therapy in medical material engineering differs from targeted therapy in oncology. In oncology, targeted therapy involves ligands that inhibit tumors by binding to specific targets, regulating corresponding pathways. These ‘targets’ typically refer to proteins or other biomolecules such as DNA, RNA, heparin, and peptides [39]. In medical material engineering, targeted therapy is categorized into two approaches based on how materials accumulate in tumor cells: active targeting and passive targeting. Passive targeting primarily relies on the enhanced permeability and retention (EPR) effect of tumor cells. This mechanism capitalizes on the rapid growth of cancer cells, characterized by leaky, layered blood vessel structures, and reduced lymphatic clearance within the tumor tissue. These factors collectively increase the penetration of circulating nanoparticles into the tumor environment, facilitating targeted drug delivery to tumor cells. However, active targeting involves combining drug carriers with receptors, peptides, or surface molecules that are highly expressed in tumor cells. This modification ensures that drug accumulation within the tumor cells exceeds that in normal tissues, achieving precise targeted drug delivery [40,41] (Figure 4). However, relying solely on either passive targeting or active targeting to increase PTA accumulation has its limitations. Currently, the primary strategy involves leveraging the advantages of both passive and active targeting to mitigate constraints and enhance the efficacy of PTT.

### 3.1. Passive Targeting

Passive targeting primarily depends on the enhanced permeability and retention (EPR) effect of tumor cells. However, numerous obstacles hinder the entry of PTA nanoparticles into tumor cells following systemic administration. These obstacles include the acidic environment of the tumor microenvironment (TME), clearance by the mononuclear phagocyte system (MPS), the avascular zone with limited convection, and intratumoral pressure [16,42,43]. In addition to augmenting active targeting, research has revealed that surface modifications or alterations in physical properties such as shape and size can enhance the accumulation of PTAs within tumors.

Nanoparticles with a neutral or negatively charged surface contribute to the dispersion and stability of PTAs in the biological environment, reducing protein adsorption and extending circulation time. Currently, surface modification using polyethylene glycol (PEG) is a widely employed method in research. PEG serves as a “stealth cloak” on the surface of nanoparticles, shielding them from aggregation, opsonization, and phagocytosis [44]. Researchers such as Klibanov and colleagues have demonstrated that PEGylation significantly increases the blood circulation half-life of systemically administered liposomes, extending it from less than 30 min to up to 5 h [45]. Other studies have indicated that the density of PEG influences the circulation time of nanoparticles [46]. To enhance the PTT effect, PEG exhibits a favorable photothermal effect on PTAs such as graphene oxide [47], MoS2 [48], and gold nanoparticles [49]. However, the modification of nanoparticles with PEG may weaken their interaction with cancer cells [16]. Consequently, the controlled release of nanoparticles, toggling between “on” and “off”, has become a focal point in research regarding novel nanomaterials.

In the context of negative targeting, coupling certain small molecules with high yet nonspecific uptake by tumor cells is a viable approach by which to amplify the EPR effect. In a study conducted by Cheng, Xu et al. [50], indocyanine green (ICG), polyacrylic acid (PAA), and n-HA were modified using glucosamine (GA) to create an organic–inorganic hybrid nanosystem referred to as GA@HAP/ICG-nps. This nanosystem capitalizes on the fact that tumor cells exhibit high glucose uptake, leading to the EPR effect within tumor tissues. Consequently, after intravenous injection, GA@HAP/ICG-NPs accumulate selectively in tumor tissues. Upon NIR irradiation, the nanosystem induced both PTT and PDT via ICG activation, achieving an impressive tumor inhibition rate of 87.89%. While the nanosystem’s osteogenic properties were not confirmed, the study introduced a novel targeting strategy for inhibiting tumors.

### 3.2. Active Targeting

At present, the technology for preparing drug-loaded nanoparticles is relatively advanced, and the combination of nonspecific tumor targeting with PTT has been extensively studied. Some of these targeting approaches also demonstrate inhibitory effects on osteosarcoma cells, for example, folic acid (FA), mitochondria-targeting, and so on (Table 1).

FA has the unique ability to selectively attach to the folate receptor found on the surface of various human tumor cells including osteosarcoma [51,52]. Xiangtian Deng et al. [53] designed and developed a bovine serum albumin-folate acid (BSA-FA) functionalized hybrid nanoplatform (IrO_2_@ZIF-8/BSA-FA (Ce_6_), denoted as IZBFC). The zeolitic imidazolate framework-8 (ZIF-8) was the drug delivery nanoplatform, IrO_2_ and Ce_6_ achieved synergistic therapeutic effects with respect to PTT-PDT, and BSA-FA on the surface of IrO_2_@ZIF-8 was the active targeting agent. The release of Ce6 increased to 42.7% at pH 5.0; in addition, IZBFC is safe and has great potential for drug delivery. Furthermore, the photothermal conversion efficiency of the IZBF NPs was determined to be approximately 62.1% in vitro and in vivo, confirming that IZBF NPs can enhance the antitumor therapeutic efficacy of PTT and PDT. FA-Fe_2_O_3_@PDA-miRNA [54] also uses FA to target osteosarcoma cells and plays a dual role in gene therapy (GT), and PTT.miR-520a-3p has the ability to expedite apoptosis in osteosarcoma cells by influencing the downregulation of the recombinant interleukin 6 receptor (IL6R) through the Jak-stat signal transduction pathway. FA-Fe_2_O_3_@PDA-miRNA can enhance the stability of miR-520a-3p. The results indicate that FA-Fe_2_O_3_@PDA-miRNA could induce satisfactory anticancer effects in osteosarcoma, and the curative ratio is better than that used alone in PTT or GT.

**Table 1 ijms-25-04139-t001:** Summary of active targets.

Author	Material	PTA	Carrier	Target
Xiangtian Deng et al. [53]	IrO_2_@ZIF-8/BSA-FA (Ce_6_)	IrO_2_ and Ce_6_	Zeolitic imidazolate framework-8 (metal-organic framework, MOF)	BSA-FA
Xue Li et al. [54]	FA-Fe_2_O_3_@PDA-miRNA	Fe_2_O_3_@PDA	Fe_2_O_3_	FA
Hongzhi Hu et al. [55]	AIBI@H-mMnO_2_-TPP@PDA-RGD(AHTPR)	MnO_2_	Hollow mesoporous MnO_2_	TPP
Wei-Nan Zeng et al. [56]	TPP-PPG@ICG	ICG	Polyethylenimine-modified PEGylated nanographene oxide sheets	TPP
Peng Lin et al. [57]	T-ND	Cy7(heptamethine cyanine)	TCF(2-dicyanomethylene-3-cyano-4,5,5-trimethyl-2,5-dihydrofuran)	OTP
Ying Yuan et al. [58]	SPN-PT	Semiconducting polymer (PCPDTBT)	PCPDTBT	Peptide PT (the original peptide that has undergone PEGylation)
Tian, J. et al. [59]	HGNs-PEG-CD271	HGNs	HGNs	CD271
Xiong, S. et al. [60]	GTN-CD133@ICG@HA	ICG	GTNs	HA, CD133
Jingwei Zhang et al. [61]	CM/SLN/ICG	ICG	SLNs (silica nanoparticles)	CMs (cell membranes)
Yanlong Xu et al. [62]	BPQDs-DOX@OPM	Black phosphorus quantum dots (BPQDs)	Black phosphorus quantum dots (BPQDs)	OPM (surface-encapsulated platelet-osteosarcoma hybrid membrane)

Triphenylphosphonium (TPP) has mitochondria-targeting properties. Many nanocomposite particles have been modified with TPP to achieve the active targeting of tumor cells including osteosarcoma cells. Hongzhi Hu et al. [55] utilized hollow mesoporous MnO_2_ (H-mMnO_2_) nanostructures as a drug delivery system to develop AIBI@H-mMnO_2_-TPP@PDA-RGD(AHTPR). Polydopamine (PDA) blocks the pores of H-mMnO_2_ and can release encapsulated drugs in an acid environment; azo initiator 2,2’-azobis [2-(2-imidazolin-2-yl) propane] dihydrochloride (AIBI) decomposes rapidly into highly reactive alkyl radicals under mild heat stimulation. AHTPR can target tumor cells via intravenous injection and pH/NIR dual-responsive drug release. Furthermore, AHTPR can achieve synergistic anticancer performance via mitochondria targeting. The synergistic therapy efficacy of Hu et al.’s system was confirmed by effectively inducing cancer cell death in vitro and completely eradicating tumors in vivo. There are also reports that TPP can be conjugated to indocyanine green (ICG)-loaded, polyethylenimine-modified PEGylated nanographene oxide sheets (TPP-PPG@ICG) to promote mitochondrial accumulation after cellular internalization [56]. TPP-PPG@ICG has exhibited a remarkably selective anticancer effect both in vitro and in vivo. This has led to the significant suppression of tumor growth in mice with doxorubicin-resistant MG63 tumor cells, with no observable toxicity, and holds significant promise for addressing drug-resistant osteosarcoma.

OS targeting peptide (OTP) is a novel strategy that can rapidly identify patient-personalized targeting ligands for osteosarcoma. Peng Lin et al. used a patient-personalized OTP that was identified via phage display techniques. A Cy7-TCF supramolecular 2D nanodisc demonstrated passive tumor-targeting properties against generic solid tumors, merely relying on EPR effects for its tumor localization. Peng Lin et al. [57] successfully conjugated OTP to Cy7-TCF. After intravenously injecting tumor-bearing mice with a single dose, they found that T-ND could not only function as a tumor-customized PTA to precisely destroy the tumor and inhibit tumor growth, but also effectively penetrate deep into tumor tissues with an ultralong tumor retention of up to 24 days. This indicates that OTP enables the active targeting of a Cy7-TCF supramolecular 2D nanodisc. Peptide PT (PPSHTPT) [63] mimics the natural protein osteocalcin property in vivo for active bone targeting. Ying Yuan et al. [58] reported on oligopeptide PT-based semiconducting polymer nanoparticles (SPN-PT). SPN-PT was formulated via the nanoprecipitation method using a semiconducting polymer (PCPDTBT) encapsuled with polyethylene glycolylated (PEGylated) PT. As a PTA, PCPDTB can be internalized into OS cells in an active fashion with PT peptide, permitting accurate early diagnosis for OS by NIR-II fluorescence. In particular, the cellular uptake of the peptide driven nanoparticles into targeted OS cells was far more rapid than the control, and it finished within 4 h.

Osteosarcoma (OS) exhibits widespread genomic alterations with few recurrent mutations. Identifying precise targets for bone tumor cells is of paramount importance in the context of targeted drug therapy. Currently, the development of novel and feasible nanomaterials for the targeted photothermal ablation of osteosarcoma stem cells is a prominent research focus. In recent years, there have been emerging specific targets designed to target osteosarcoma while harnessing the photothermal effect to inhibit osteosarcoma cells. Examples include CD271, CD133, and the osteosarcoma targeting peptide (OTP). CD271 is considered to be a surface biomarker for osteosarcoma stem cells. Tian, J. et al. [59] developed PEGylated multifunctional hollow gold nanospheres (HGNs) based on the CD271 monoclonal antibody. Bifunctional SHPEG-COOH was used to facilitate the covalent linkage between HGNs and the CD127 antibody (HGNs-PEG-CD271). HGNs-PEG-CD271 achieved excellent cell viability inhibition with NIR. The targeting of CD271 was confirmed. In addition, HGNs are hollow nanostructures that can be used as carriers to deliver drugs. CD133 is a key marker for screening tumor stem cells Xiong, S. et al. [60] also constructed GTN-CD133@ICG@HA using gold nanoparticles loaded with CD133, ICG, and hyaluronic acid A(HA). CD133 targeted osteosarcoma tumor stem cells and HA targeted tumor cells and protected the photosensitized drugs loaded onto nanoprobes. Furthermore, when induced by mild irradiation with a single-wavelength laser, this treatment effectively inhibited tumor growth in an osteosarcoma mouse model.

Nowadays, biofilm systems are widely used because they are not cleared by the immune system, exhibit noncytotoxicity and high bioavailability, and inherit the merits of the source cells [64,65,66]. Various cell membranes (CMs) have been utilized in the development of bioinspired drug delivery systems (DDSs) for the targeted therapy of diseases [67,68]. Jingwei Zhang et al. [61] utilized a cell membrane (CM) derived from 143B cells to target homogenous 143B cells. They surface modified silica nanoparticles (SLNs) with CM and encapsulated indocyanine green (ICG), which is a PTA, to construct a platform (CM/SLN/ICG). CM/SLN/ICG demonstrated the specific targeting of homogenous 143B cells in both in vitro and in vivo experiments. This resulted in superior anticancer efficacy when compared to either SLN/ICG or free ICG. Yanlong Xu et al. [62] employed BPQDs-DOX@OPM. Black phosphorus quantum dots (BPQDs), a PS known for its high drug loading capacity, were used to load DOX. Subsequently, the researchers utilized a surface-encapsulated platelet-osteosarcoma hybrid membrane (OPM) to encapsulate the surface of BPQDs-DOX, thereby extending the DOX circulation time and enabling OS isotype targeting. The combined therapy using BPQDs-DOX@OPM had several advantages compared to single-agent chemotherapy including prolonged circulation time, targeted drug delivery, enhanced antitumor activity, and a high level of biosafety.

## 4. PTT Combined Multiple Therapies

In current malignant tumor treatment protocols, combination therapy has emerged as the predominant approach, with PTT representing a promising treatment modality that can be integrated with various other therapeutic strategies to combat tumors effectively. Two primary forms of combined therapy exist: firstly, PTT serves as an adjunct to complement other treatment modalities such as post-tumor surgery PTT aimed at reducing tumor recurrence rates [69,70]; secondly, altering the structure of the PTA or coupling it with molecules possessing antitumor properties facilitates the development of materials capable of not only executing PTT, but also synergizing with additional therapeutic methodologies for tumor management. PTT predominantly operates via PTA mechanisms. Presently, ongoing innovation in nanomaterials enables modifications to PTA structures, facilitating their integration with chemical, immunological, and chemical kinetic treatment approaches (Figure 5), thereby enhancing the overall therapeutic outcomes.

### 4.1. PTT Combined with Chemotherapies

In the context of inhibiting tumor recurrence, chemotherapy represents the most common clinical treatment approach. However, due to its nonspecific impact on normal cells, it often leads to systemic complications and toxic side effects including nausea, vomiting, alopecia, and compromised immunity [71].

The utilization of nanoparticles as drug carriers for transporting and releasing chemotherapeutic agents has garnered extensive attention [40,72,73]. This approach mitigates the complications and toxic side effects associated with systemic chemotherapy. Beyond the advantages mentioned earlier, some nanomaterials combine with biological molecules or residues to heighten the specificity of chemical drug complexes in targeted therapy, thus enhancing the efficacy of nanomaterial-based treatments [35,73,74]. Combining phototherapy with antitumor drugs using nanoparticles allows for the targeted and synergistic eradication of cancer cells, addressing the limitations of single treatment modalities. This approach holds significant promise for development and application. Nonetheless, this treatment strategy still has inherent issues such as drug side effects, sustained release, and drug resistance, which can impact its efficacy in tumor treatment [75,76,77].

Curcumin (CM) is a polyphenolic compound known for its antioxidative, anti-inflammatory, and anticancer properties [78,79]. Because of its mild pharmacological effects, it is deemed safe for human consumption [80]. It has been extensively studied in cancer treatment, particularly in the context of osteosarcoma [81,82]. Its mechanisms of action primarily involve inducing apoptosis, inhibiting proliferation and metastasis, increasing cytotoxicity, and promoting cell cycle arrest. Specifically, curcumin can exert its destructive effects on osteosarcoma cells by modulating the activity of various signaling pathways such as HMOX1 [83], JAK-STAT [84], and Smad as well as regulating microRNA and gene expression [85]. These mechanisms confer curcumin with potential anticancer effects, offering new directions and strategies for osteosarcoma treatment. X. Sun et al. synthesized CM-CS nanoparticles [86] and CM-PDA/SF/n-HA nanofibers [87]. When MG-63 cells were co-cultured with CM-PDA/SF/1%n-HA and exposed to NIR irradiation, the cell survival rate dropped from 60% to 20%. The scaffold’s elevated temperature under NIR irradiation directly hindered tumor cell growth and enhanced CM drug penetration, promoting effective drug release. Additionally, the incorporation of SF and n-HA improved scaffold biocompatibility and bolstered its capacity to induce osteogenic differentiation and facilitate new bone formation. Bowen Tan et al. [88] took a different approach by embedding CM and indocyanine green into a poly(lactic-co-glycolic acid) (PLGA) hydrogel to create an injectable formulation. In an in situ osteosarcoma model, the hydrogel +NIR group demonstrated significant local tumor clearance. Furthermore, micro-computed tomography (micro-CT) revealed a normal tibia morphology, while the non-NIR group exhibited negligible antitumor effects. This highlighted the hydrogel’s ability to maintain stable local CM concentrations, serving a dual role of photothermal synergy to inhibit tumors and promote bone repair and new bone formation upon illumination.

Furthermore, certain materials can serve as carriers for chemotherapy drugs, enabling both photothermal effects and precise drug delivery triggered by light exposure. Core-shell nanostructures have been employed for this purpose, particularly in the delivery of the drug doxorubicin hydrochloride (DOX) [89]. These nanostructures consist of a core comprising of mesoporous silica, while the shell features a sandwich structure incorporating Hypocrellin A (HA)/carbon dots (CDs) and a mesoporous silica layer. DOX is loaded into the mesopores through a photo-unstable agent known as the o-nitrobenzyl derivative linker (NB linker). The particle’s entire surface is modified with lactobionic acid (LA), which binds to LA receptors that are highly expressed by tumor cells. This modification ensures targeted drug delivery to tumor cells following intravenous injection. Upon exposure to 980 nm light, DOX is released, while Hypocrellin A and the carbon dots contribute to phototherapy, effectively inhibiting tumor cells under near-infrared (NIR) irradiation.

### 4.2. PTT Combined with Immunotherapy: PTT—Immunomodulation

The ability to evade immune system surveillance and passivate immunogenicity is the primary reason for the occurrence and development of tumors [90]. In the process of tumor elimination, the acute inflammatory response triggered by tumor-associated antigens (TAAs) induces the activation of dendritic cells (DCs). Then, upon activation, DCs present tumor antigens and activate tumor-specific CD8+ cytotoxic T lymphocytes (CTLs) to kill tumor cells. Tumor cells that can reduce their immunogenicity through immunoediting survive or produce certain negative regulators including PD-L1 on tumor cells, interleukin-10 (IL-10), transforming growth factor β (TGF-β), regulatory T cells (Tregs), and myeloid-derived suppressor cells (MDSCs) in the tumor microenvironment (TME). Consequently, immune escape occurs [91,92,93].

Recently, immunotherapy in which the body’s immune system is trained to recognize and fight against tumors has shown great potential for cancer treatment [94], especially for aggressive and metastatic tumors. Cancer immunotherapy encompasses several approaches including tumor vaccines, immune checkpoint blockade (ICB), and chimeric antigen receptor T cell (CAR-T) therapy.

However, cancer immunotherapy faces several limitations: (1) patients exhibit significant individual differences in treatment responses; (2) the therapy can be costly and may lead to cytokine storms [95]; (3) the low immunogenicity of tumor cells and the immunosuppressive tumor microenvironment can restrict treatment efficacy [96]; (4) tumor vaccines targeting a single tumor antigen are less effective than those targeting multiple tumor antigens and are rarely suitable as universal vaccines [97,98]; (5) only a small percentage of cancer patients (10–30%) benefit from current immune checkpoint inhibitor (ICI) treatments, and long-term ICI use can lead to immune-related side effects [99,100].

Given these limitations, there is a pressing need to develop treatments that not only effectively eliminate tumors, but also reduce the risks of metastasis and recurrence. Such treatments should be less impacted by individual patient differences. One of the current research frontiers addresses this need through the combination of PTT with immunotherapy. As previously mentioned, PTAs are capable of converting light energy into heat energy, resulting in a high photothermal conversion rate. This effect, when induced by irradiation, leads to the denaturation of heat shock proteins within tumor cells, chromatin alterations, and the inhibition of DNA synthesis and repair, ultimately culminating in tumor cell death due to the elevated temperature. PTT also triggers the production of tumor-specific cytotoxic T lymphocytes (CTLs) and significantly increases the expression of costimulatory molecules (CD40, CD80, CD86) on dendritic cells (DCs), promoting the secretion of interleukin (IL)-12 and activating adaptive immunity [101,102,103]. Recent research has further explored the mechanisms underpinning PTT’s inhibitory effect on tumor cells. PTT enhances tumor immunogenicity by inducing immunogenic cell death (ICD) [104] and recruiting endogenous cytotoxic T lymphocytes (CTLs) [105]. ICD can lead to the release of damage-associated molecular patterns (DAMPs) and tumor-associated antigens (TAAs), bolstering innate immunity. Additionally, nano-PTA is readily absorbed by DCs following tumor cell destruction, facilitating antigen presentation by DCs and promoting the specific immune targeting of tumor cells [106]. However, it is important to note that the immunosuppressive tumor microenvironment (TME) can significantly impede ICD-driven immunity. Consequently, immunotherapy triggered solely by PTT may prove insufficient. This approach often falls short in achieving sustained, long-term tumor remission and carries a heightened risk of tumor recurrence and metastasis [107].

In the investigation of PTT’s mechanism for combating osteosarcoma, there have been reports indicating that PTT can additionally modulate the growth and invasive potential of osteosarcoma tumors by influencing macrophage polarization. Tumor-associated macrophages (TAMs) are macrophages infiltrating into tumor tissues and are the most abundant immune cells in the tumor microenvironment [108]. The proportion of TAMs in osteosarcoma accounts for more than 50% of the immune cell content, and TAMs may participate in the malignant progression of osteosarcoma [109]. TAMs have two subtypes of macrophages: M1 (induced by lipopolysaccharide and interferon-γ) and M2 (induced by interleukin-4 and interleukin-13) [110]. M1 macrophages (activated macrophages) have antibacterial and antitumor effects, whereas M2 macrophages have the opposite effect and contribute the most to the proportion of TAMs [111]. To explore whether PTT can modulate TAMs to exert antitumor effects, in 2020, Xiangyu Deng et al. [112] developed GO/PEG, a photothermal material that induces a heating effect in macrophages, by combining graphene oxide (GO) with polyethylene glycol (PEG). RAW264.7 was cultured with GO/PEG. GO/PEG could alleviate the interleukin-4-induced M2 polarization of macrophages and modulate their antitumor capability with NIR. The supernatant was cultured with HOS cells and injected into BALB/C nude mice. The migration and invasion capabilities of HOS cells were weakened, leading to an antitumor effect in vivo. The results suggest that the cell supernatants of macrophages treated with PTT can help inhibit the growth and invasion ability of tumors in osteosarcoma. There are also scientists reprogramming macrophages to improve the outcome of OS. The authors of [113] developed a biodegradable material based on hydroxypropyl chitin (HPCH), tannic acid, and ferric ions (HTA) in the form of an injectable and photothermal hydrogel, which was designed to reprogram TAMs into classically activated macrophages (M1). HTA + NIR led to photothermal tumor cell destruction in vitro and vivo. Therefore, targeting TAMs as a complementary therapy is expected to improve the prognosis of osteosarcoma.

PTT has the potential to enhance tumor immunogenicity and disrupt the immunosuppressive nature of the tumor microenvironment, thereby augmenting the effectiveness of standalone immunotherapy. Conversely, immunotherapy can establish long-lasting immune memory, reducing the risk of tumor metastasis and recurrence. Combining PTT with immunotherapy capitalizes on the strengths of both approaches while compensating for their individual limitations. This novel approach is referred to as PTT-immunomodulation and can be implemented through four primary binding methods: (1) nano-PTT combined with antigens; (2) combining with immune adjuvants to create an in situ vaccine; (3) combining with CAR-T therapy; (4) integrating with checkpoint inhibitors [107,114].

The effectiveness of photodynamic immunotherapy has been demonstrated in inhibiting both in situ tumors and metastatic tumors using localized light in conjunction with immunotherapy. This approach offers noninvasive and targeted tumor metastasis inhibition as well as systemic treatment in 4T1 cells and B16F10 cells [115]. Osteosarcoma is characterized by its high metastatic potential, which poses a significant challenge in terms of tumor recurrence and progression [116]. Despite ongoing research, immunotherapy remains the most promising treatment strategy for improving the currently stagnant 5-year survival rate of patients [117]. PTT, in conjunction with immunotherapy, offers a potential solution to enhance the effectiveness of current immunotherapies. Nanophotosensitizers, when activated for PTT, can trigger the maturation of dendritic cells (DCs) through specific pathways. This process not only augments the innate immunity within tumors, but also enhances the immunotherapeutic effects of the immune checkpoint blockade (ICB) or immune adjuvants when combined with nano-PTT. The research focus now centers on elucidating the mechanisms by which PTT inhibits osteosarcoma, optimizing the synergy between nano-PTA and immune drugs, and developing effective strategies for targeting bone tumor cells following systemic administration. These investigations hold promise for advancing osteosarcoma treatment.

The intricate tumor microenvironment of osteosarcoma presents a challenging landscape, and the precise mechanisms underlying PTT are still the subject of exploration. Limited research has ventured into leveraging the multifunctional capabilities of nano-PTA such as its potential as a carrier and targeting agent in combination with immunotherapy. Currently, the most prevalent approach involves modifying nano-PTA with various functional molecules to elicit the transformation of osteosarcoma into “hot” tumors or to stimulate dendritic cell (DC) maturation. This combination therapy aims to enhance the immune response against osteosarcoma cells while inhibiting their growth. Further investigations in this direction hold promise for advancing our understanding and treatment of osteosarcoma.

Guan Ping He [118] and colleagues developed AuNDs@aPD-1, with AuNDs as the PTA and the programmed receptor 1 antibody (aPD-1) as the immune checkpoint inhibitor (ICI). By combining passive targeting through the enhanced permeability and retention (EPR) effect with active targeting using aPD-1, they aimed to increase the accumulation of these agents in tumor cells while reducing systemic side effects related to ICIs. The study involved creating an orthotopic osteosarcoma (OS) model in the proximal tibia and an axillary OS metastasis model in mice. Intravenous systemic administration was carried out, and near-infrared (NIR) treatment was applied solely to the tibial tumor. Subsequently, an axillary OS model was established again to assess long-term tumor memory function. The results support the notion that the strategic combination of PTAs and ICIs enhances the synergistic antitumor effect, effectively suppressing primary, distant, and metastatic tumors, extending survival, and preventing tumor recurrence. It was observed that PTT induced immunogenic cell death (ICD), releasing damage-associated molecular patterns (DAMPs) and tumor-associated antigens (TAAs), with mt-DNA in DAMPs sensed by cyclic GMP-AMP synthase (cGAS). This activated the cGAS-STING pathway, a crucial route linked to DC maturation [119,120]. However, single PTT had limitations in inducing innate and specific immunity. Kaiyuan Liu et al. [121] proposed the design of a material that could not only enhance the immunogenicity triggered by PTT, but also elevate tumor-specific immunity. They created a nanoplatform using titanium carbide MXene loaded with Mn^2+^ and ovalbumin (OVA). OVA derived from tumor cells enhanced adaptive immunity, while the combination of titanium carbide MXene and Mn^2+^ synergistically activated innate immunity through processes involving immunogenic cell death (ICD) and mitochondrial DNA (mt-DNA). The results indicate that this nanoplatform effectively facilitated the presentation of tumor antigens and significantly promoted the maturation of dendritic cells (DCs). This enhancement resulted in the improved infiltration of cytotoxic T lymphocytes into both primary and distant tumors.

### 4.3. PTT Combined with Chemodynamic Therapy—Photothermal-Chemodynamic

Chemodynamic therapy (CDT) is an emerging nanocatalyst-based therapeutic strategy for tumor-specific therapy [122]. CDT is considered to be a powerful type of reactive oxygen species (ROS)-based antitumor therapy [123]. CDT primarily relies on the Fenton reaction or Fenton-like reactions. In these processes, hydrogen peroxide (H_2_O_2_) present in the acidic tumor microenvironment undergoes catalysis by nanometal ions. This catalytic reaction generates a substantial amount of toxic reactive oxygen species (ROS), effectively targeting and destroying malignant cells [124,125].

As a novel tumor treatment, CDT has undergone thorough investigation. However, there are still some challenges with respect to its application in vivo. An insufficient Fenton reaction yields a limited amount of reactive oxygen species (ROS), which proves inadequate in effectively inhibiting tumor cells [122,126]. Elevating the temperature can promote Fenton or Fenton-like reactions, thereby releasing more ROS and augmenting the antitumor efficacy of catalytic therapy (CDT) [127]. PTT also elevates the local tumor temperature for inhibiting tumor cells. Consequently, combining PTT with CDT is a viable strategy by which to create a synergistic antitumor effect. Hence, there is a pressing demand for the development of multifunctional nanomaterials possessing both photothermal (PT) and chemodynamic properties for enhanced PTT-integrated catalytic therapy (CDT). This is called the photothermal-chemodynamic approach and holds promise for advancing the field of combination cancer therapies, with the aim to achieve highly effective treatment modalities.

The limited studies on multifunctional metal nanoparticles combining PTT with catalytic therapy (CDT) in osteosarcoma are attributed to the challenges in constructing an osteosarcoma tumor model. Moreover, the majority of the existing research has predominantly employed 4T1 tumor cells. In the realm of metal-based nano-ions, their loading onto a PTA for a synergistic PTT/CDT effect is a widely explored avenue. Notably, materials such as polydopamine (PDA), porous carbons, metal–organic frameworks, and transition metal carbide (MXene) feature prominently in these studies [128]. PDA nanomaterials have been widely explored as carriers to enhance the drug inhibition of tumor cells under near-infrared (NIR) irradiation [129]. In an innovative study by Qian Chen et al. [130], PDA was combined with tannic acid-Fe^3+^ (TA-Fe) to create PDA@TA-Fe nanoparticles. TA-Fe exhibits an enhanced permeability and retention (EPR) effect in tumor cells, further boosted by the NIR-induced production of cytotoxic hydroxyl radicals via the Fenton reaction. Together, these elements synergistically inhibit 4T1 tumor cells. Furthermore, enhancing PTT and catalytic therapy (CDT) can be achieved by loading nanocellular-based carriers with highly catalytic enzymes possessing PTT properties. In a study by Yanling Liang et al. [131], hybrid bimetallic RhRu/Ti_3_C_2_Tx nanozymes with high catalytic efficiency were anchored onto 2D Ti_3_C_2_Tx nanosheets. This innovative approach effectively combines the benefits of PTT and CDT.

### 4.4. PTT Combined with Magnetic Hyperthermia—Magnetic Photothermal Therapy (MPHT)

Magnetic hyperthermia (MH) represents a noninvasive, localized hyperthermia therapy method by applying a high-frequency alternating magnetic field (AMF) to magnetic nanoparticles (MNPs), resulting in localized heat generation (42–45 °C) that heats the tumor tissue and induces the thermosensitive death of cancer cells. It is a novel method of localized hyperthermia therapy that is noninvasive. MNPs are commonly synthesized using pure metals like iron (Fe), cobalt (Co), nickel (Ni), and certain rare earth metals, or through blending metals with polymers. However, due to the instability and toxicity of pure metals in the human body, metal oxides are often selected as MNPs [132]. The primary advantage of this method lies in its use of localized hyperthermia, leveraging the higher sensitivity of tumor cells to temperature compared to normal cells, thereby minimizing its impact on healthy tissues. Additionally, we can manipulate MNPs by artificially controlling the AMF. These MNPs utilize temperature to suppress tumor cells, but recent studies have also suggested that magnetic fields can stimulate osteoblast proliferation and differentiation, promote BMP expression, and accelerate new bone formation [133,134].

In clinical applications, the main critical factor of MH is the thermal conversion efficiency of MNPs described by the specific absorption rate (SAR) [135]. However, in the research with respect to MH, it has been observed that certain MNPs exhibit an uneven distribution within tumors, leading to areas with excessively high or low temperatures, causing uneven tumor suppression [136]. Researchers can modify the chemical structure (surface coatings, targeting agents) and general physical properties (size, shape) of MNPs to adjust the SAR, thereby improving the thermal conversion efficiency of MNPs and augmenting their anticancer effects. Tong et al. found, through a refined dynamic hysteresis model, that the thermal efficiency of magnetic iron oxide nanoparticles (MIONs) increased with particle size [137]. Coatings such as PEG and GO reduce toxicity and nanoparticle aggregation, significantly enhancing their SAR [138]. Targeting agent (peptides, ligands, lipids, aptamers) conjugation can increase accumulation in target tumor cells and improve the tumor cell suppression efficiency [139]. Yu et al. incorporated glucose oxidase (GOx), MgCO_3_, and Fe_3_O_4_ into a PLGA hydrogel to synthesize a triple-functional magnetic gel (Fe_3_O_4_/GOx/MgCO_3_@PLGA) [140]. Under an AMF, this gel induces MH therapy, where GOx is released to inhibit ATP production, thereby reducing HSP expression. This synergistically inhibits the growth of osteosarcoma cells while releasing magnesium ions to promote bone regeneration at defect sites.

However, the temperature generated by MH remains relatively low and research has shown an upregulation of heat shock protein (HSP) expression during MH, increasing the intrinsic heat resistance of tumor cells [141,142]. Therefore, the effectiveness of standalone MH is ultimately limited. Researchers have proposed combining MH with other therapies to enhance its anticancer capabilities or achieve multifunctional objectives. MH can be combined with chemotherapy [143,144], immunotherapy [145], and radiotherapy [146], while PTT can also be considered a form of thermal therapy. Recent advancements in biomedical material research have revealed that certain MNPs also possess the property of converting light energy into heat. This combination approach is referred to as magnetic photothermal therapy (MPHT).

MPHT has been extensively studied, prompted by the limited tissue penetration of PTT and the shallower depth constraints of MH. Upon combining PTT with MH, NIR irradiation after the application of AMF can increase the thermal conversion efficiency of MNPs, thus, augmenting the suppression of tumor cells. This combination has the potential to mutually alleviate the limitations of each method and achieve the synergistic suppression of tumor growth. In a study conducted by Das et al., Fe_3_O_4_ nanoparticles were structured in a flower-like arrangement surrounding Ag cores. Upon simultaneous exposure to an external magnetic field and laser irradiation, the SAR of the nanoflowers significantly increased by at least an order of magnitude, with decreased demand for magnetic field and laser intensity in comparison to individual stimuli [147]. Similarly, Alberto Curcio and colleagues synthesized spiky IONF@CuS nanohybrids, employing magnetite (γ-Fe_2_O_3_) and CuS as PTAs. Their findings demonstrate that the heating capacity of PTT exceeded that of MH at equivalent concentrations, suggesting cumulative effects of both heating modalities. In vitro and in vivo experiments conducted on PC3 cells validated that the combined application of MH and PTT synergistically eradicated tumors and demonstrated superior efficacy compared to MH monotherapy [148]. Açelya Yilmazer et al. illustrated the synergistic anticancer effects of MH and PTT and observed elevated levels of calreticulin expression, extracellular ATP, HMGB1, dendritic cell and natural killer cell aggregation, and ferroptosis following MPHT, implying connections with ICD and ferroptosis [149].

In addition to the amalgamation of PTT and MH, they can also be integrated with chemotherapy to exert triple-functional effects. The utilization of magnetic nanoparticles as drug carriers for combined MH and chemotherapy has been documented. Whether nanoparticles possessing both PTT and MH capabilities as drug carriers can achieve triple functionality remains under investigation. Martínez-Banderas and colleagues employed nanowires with an iron core and iron oxide shell as carriers for doxorubicin (DOX) [150]. Treatment with DOX under simultaneous exposure to AFM and NIR radiation led to a 91% reduction in cell viability. Additionally, there was a 30% increase in cytotoxicity under NIR alone and a 15% increase under AFM alone, indicating greater toxicity compared to DOX release alone. Furthermore, the internalization of nanowires facilitated selective DOX release within the nucleus, potentially mitigating DOX’s adverse effects.

PTT can also be integrated with MNPs to concurrently promote bone regeneration and inhibit tumor cells. Considering the challenges encountered by patients with bone tumors in postoperative bone defect repair and residual tumor elimination, PTT can serve as a tool for eliminating residual tumors. As previously mentioned, MNPs possess the capability to promote osteogenic differentiation, thereby warranting further investigation. Jia-Wei Lu and colleagues modified a scaffold composed of bioglass (BG) and chitosan (CS) with SrFe_12_O_19_ magnetic nanoparticles [151]. SrFe_12_O_19_ exhibited outstanding magnetism and served as a photothermal agent (PTA) for PTT. The researchers showed that the magnetic field produced by the modified scaffold not only accelerated the proliferation of human bone marrow-derived mesenchymal stem cells (hBMSCs), but also markedly upregulated the expression levels of osteogenesis-related genes (OCN, COL1, Runx2, ALP). Upon exposure to near-infrared laser irradiation (808 nm, 4.6 W·cm^−2^) for 2 min, the scaffold quickly reached 43 °C, demonstrating its PTT effect. The modified scaffold demonstrated remarkable properties for bone regeneration and exceptional functionality for MPHT.

### 4.5. PTT Combined with PDT

The combination of PTT and PDT, both being types of phototherapy, relies on light irradiation to exert their antitumor effects. However, using either method alone has limitations. PTT combined with PDT is deemed more effective compared to their singular use. PDT has been shown to enhance the sensitivity of tumor cells to temperature, thereby augmenting the efficacy of PTT [152]. Additionally, PTT can increase intra-tumoral blood flow and enhance the oxygen environment within the tumor, facilitating PDT-induced ROS production for tumor cell destruction and function, thus reducing the dosage of photosensitizers and related side effects [153,154].

In the context of bone tumor treatment, the combination of PTT and PDT has been relatively less explored, primarily due to the disparate parameters such as wavelength and frequency required for their excitation. While PTT primarily relies on NIR light, PDT utilizes ultraviolet or visible light sources. Thus, further exploration is warranted to determine how to optimize the efficiency of both methods.

In the pursuit of substances with dual functionality, it has been shown that some noble metal nanoenzymes exhibit catalase-like activity (CAT), facilitating the catalysis of hydrogen peroxide to generate singlet oxygen under NIR irradiation, thereby enabling PDT. Conversely, metal oxides can serve as PTAs under NIR, inducing PTT effects [155,156]. Despite this, the catalytic capacity of single metal oxide enzymes remains limited. In the context of bone tumor treatment research, Yanling Liang and colleagues successfully enhanced the catalytic activity of nanoenzymes by loading RhRu bimetallic hybrid nanoenzymes onto 2D Ti_3_C_2_Tx nanosheets, yielding RhRu/Ti_3_C_2_Tx. In vitro studies and nude mice with subcutaneous osteosarcoma models demonstrated the advantages of combined CDT/PDT/PTT therapy under NIR (808 nm) irradiation [131]. Additionally, bismuth (Bi)-based nanomedicines exhibit dual functionality in PTT and PDT. Jing Cheng et al. synthesized AgBiS_2_ nanoparticles, achieving a high photothermal conversion efficiency of 36.51% under NIR, significantly increasing ROS production. Moreover, the presence of silver (Ag) and bismuth (Bi) allows this material to function as a contrast agent for CT imaging and possess certain antibacterial properties, thereby preventing Staphylococcus aureus infections.

Recent research has shown that lanthanide ion-doped upconversion nanoparticles (UCNPs) can convert near-infrared light required for PTT into high-energy visible or ultraviolet light to activate PDT [157]. Zhaoyou Chu et al. constructed core–shell structured UCNPs@AgBiS_2_ by combining UCNPs with AgBiS_2_ [158]. By doping different concentrations of Nd ions, the photothermal conversion efficiency was increased from 14.7% to 45%. The research also demonstrated that the production of ROS under 808 nm near-infrared light was significantly higher than that of pure AgBiS_2_. The mechanism behind this may involve continuous Nd^3+^ → Yb^3+^ → activator energy transfer, activating AgBiS2 to generate ROS through energy transfer. Xueyuan Guo et al. assembled two polymers with UCNPs to form UCNP-CPs. Among them, PPVBT-COOH exhibited excellent PDT effects, while PDTPBT-COOH showed excellent PTT effects. Only UCNPs-CPs demonstrated the ability to generate ROS under 980 nm near-infrared light, further emphasizing the importance of UCNPs in converting light sources for combined PDT and PTT [159].

## 5. Novel Calcium Phosphate Scaffolds—Bifunctional Scaffolds for Photothermal Therapy and Bone Regeneration

At present, the standard surgical procedure used to achieve limb salvage correction in bone tumors is surgical resection and bone defect reconstruction. The risks and causes of recurrence after tumor resection have been described above; the reconstruction of bone is of paramount importance for the patient’s quality of life and remains a prominent focus post-surgery. Bone is mainly composed of three parts: cells, fibers, and matrix. The major component of the bone matrix is collagen, which provides tensile strength. The mineral composition of bone mainly comprises calcium and phosphorus, which provide compressive strength [160]. So far, the clinical methods used for repair include autogenous bone grafting, allogeneic bone grafting, bone transport, membrane-induced osteogenesis, and artificial material reconstruction [161,162,163]. However, all have some advantages and disadvantages. Of these methods, autologous bone grafts have excellent biocompatibility, natural osseointegration, and osteoinductivity. However, the amount of autologous grafting requires alternative approaches for critical bone defects, which result in high costs and risks of disease transmission and immune rejection [163,164,165,166]. For malignant bone tumor surgery, bone grafting alone cannot meet the clinical needs due to large bone defects and poor patient tolerance. Therefore, there is a need to develop biocompatible, high mechanical strength but non-immunogenic bone repair substitute materials, namely biomaterials. The bone tissue engineering (BTE) procedure holds great promise for addressing this issue, and scaffold biomaterials are critical to its success. In 1969, Hench [167] developed bioactive glass and proposed that biological activity means that the transplanted material and biological tissue contact can be chemically combined. After the graft fills the gap and is fixed, bone mesenchymal stem cells (BMSCs) and other required cells, nutrients, and molecules are transported to the scaffold. They adhere to the scaffold, promoting cell growth and the formation of blood vessels, ultimately stimulating new bone formation [168].

As research with respect to BTE has progressed, it has been established that BTE promotes osteogenesis by creating optimal biomimetic environments that enhance the regeneration and growth of normal tissues and cells. This is achieved by combining stem cells, scaffolds, and growth factors (GFs) [169]. Firstly, BTE scaffolds must possess the appropriate pore size and porosity to support cell adhesion and facilitate nutrient exchange. Moreover, mechanical and degradation properties essential for osteogenesis play a pivotal role in the formation of new bone [170].

Scaffolds in BTE are typically classified into two categories: inorganic bioceramics and organic polymers. Both categories hold the potential for achieving optimal osteogenesis [166]. The materials used include polymers (natural or synthetic), ceramics (calcium-phosphate, bioglasses, or glass-ceramics), metals, and their composites. However, each material comes with its own set of advantages and disadvantages. Biological ceramics and organic polymers exhibit excellent biological compatibility, while metals demonstrate superior mechanical strength.

The research focus of BTE for large bone defects post bone tumor surgery include the following: (1) building scaffolds with desirable shape, structural, physical, chemical, and biological features for enhanced biological performance and for regenerating complex bone tissues [171]; (2) in situ elimination of residual tumor cells to mitigate postoperative tumor recurrence. Notably, recent studies have highlighted the integration of PTA into biological scaffolds to effectively target tumor cells and facilitate new bone formation (Figure 6).

Nano-hydroxyapatite(n-HA) and tricalcium phosphate (TCP) are widely studied biomaterials in the field of bioceramics. The composition of these scaffold material closely resembles the inorganic makeup of bone. As the scaffold degrades, it can modulate the regenerative microenvironment to accommodate bone tissue growth. Simultaneously, the mechanical properties of the graft diminish gradually, with the body’s biological stress shifting from the graft to the newly forming bone tissue. This process effectively prevents stress shielding and fosters tissue regeneration [172].

### 5.1. Tricalcium Phosphate (TCP)

TCP, Ca_3_(PO_4_)_2_, is a commonly used absorbable bioactive ceramic material with a Ca/P ratio of 1.5. Based on its crystal structure, it can be classified into high-temperature α-TCP and low-temperature β-TCP. α-TCP is formed either by heating low-temperature crystalline β-TCP or through the hot crystallization of an amorphous precursor with the appropriate composition above the phase transition temperature [173]. Despite sharing the same chemical composition, α-TCP and β-TCP differ significantly in terms of structure, density, and solubility. Consequently, their applications in the field of bone repair materials are distinct. β-TCP exhibits excellent bone conductivity, bone induction capabilities, and degradability, making it conducive to the growth of new bone around implanted TCP scaffolds [174,175]. Conversely, α-TCP is more soluble and can rapidly hydrolyze into calcium-deficient hydroxyapatite. This property makes α-TCP a valuable component in the preparation of self-setting bone conduction cement and biodegradable bioceramic materials for bone repair [173]. Studies have demonstrated the high osteoinductive potential of β-TCP [176], making it an appealing choice for bone graft applications.

Carbon-based PTAs are commonly combined with tricalcium phosphate (TCP) in composite materials including graphene oxide (GO), carbon aerogel (CA), etc. This composite material serves a dual function, acting as both a PTA for antitumor effects and TCP to promote osteogenesis. Additionally, the composite material can enhance the osteogenesis effect of TCP through near-infrared (NIR) radiation.

GO belongs to the graphene family and features various functional groups on its two-dimensional surface including carboxyl groups, hydroxyl groups, epoxy functional groups, and more. GO possesses a significant specific surface area and an abundance of surface defects, making it suitable for the loading of inorganic nanoparticles. These nanoparticles can be attached to the GO surface through in situ growth or post-modification. In 2016, Jinfeng Liao et al. [177] employed immersion drying to apply a coating of GO onto the surface of 3D-printed β-TCP scaffolds. The study confirmed that the scaffold had the capacity to stimulate new bone formation and effectively eradicate human osteosarcoma cells when exposed to near-infrared light (808 nm, NIR, 0.36 W/cm^2^, 50 °C, 10 min). These findings were established through in vitro co-culture experiments with bone marrow mesenchymal stem cells (BMSCs) as well as in vivo tests utilizing a nude mouse osteosarcoma model and a mouse skull defect model. The successful creation of this dual-function scaffold marked the beginning of numerous potential dual-function scaffolds. Researchers have conclusively demonstrated that PTA not only exhibits a photothermal effect, but also synergistically promotes bone regeneration when incorporated into composite scaffolds such as carbon aerogels (CAs) [178]. CAs are another carbon-based material known for their distinctive features including a large specific surface area, ultra-low density, and high porosity. In one study, through the infiltration drying method, a CA was successfully integrated onto a β-TCP scaffold, creating a CA coating. During co-culture with BMSCs, it was observed that cells could migrate and adhere to the pores of the TCP-CA scaffold, consequently stimulating bone cell regeneration. In a nude mouse bone defect model, it was evident that bone regeneration was significantly more pronounced when compared to the use of a TCP scaffold alone. This experiment highlights the potential for synergistic effects when combining PTA and bioactive scaffolds, capitalizing on their respective advantages.

When TCP is combined with a multifunctional PTA, it assumes the dual functions of promoting osteogenesis and inhibiting tumor cells through various mechanisms. Typically, this multifunctional PTA comprises a complex of various bioactive ions. These active ions can be released under NIR radiation to exert biological activity, promoting angiogenesis, fostering new bone formation, or inducing a PTT effect. The aim of this approach is to achieve a synergistic bifunctional effect through multiple mechanisms.

In a study conducted by Chao Xu et al. [179], Fepse3 nanosheets were successfully integrated into α-TCP. This integration serves a dual purpose: the eradication of tumor cells and the promotion of bone regeneration. On the one hand, Fepse3 acts as a semiconductor PS, demonstrating a photothermal effect when exposed to near-infrared light. Additionally, it is a degradable material capable of breaking down and releasing selenium (Se). For bone regeneration, α-TCP naturally degrades into calcium (Ca) and phosphorus (P), while the degradation of Fepse3 also results in the release of iron (Fe) and phosphorus (P). Consequently, the scaffold releases essential bioactive ions such as Fe, Ca, and P. This release, in turn, activates the expression of CD31, von Willebrand factor (vWF), and vascular endothelial growth factor (VEGF), facilitating vascularized bone regeneration and ultimately improving the scaffold’s ability to promote bone defect repair. LaB6 is a chemical compound comprising La and B elements and is a type of metal-based PTA that demonstrates excellent near-infrared photothermal efficiency [180]. Additionally, La3+ has the potential to address bone resorption disorders such as osteoporosis, while boron (B), an essential micronutrient for the human body, can promote wound healing and stimulate the release of osteoinductive growth factors [181]. Wentao Dang et al. [182] utilized PDLLA as a medium in which to synthesize TCP-PDLLA-LB scaffolds by incorporating LaB6 onto TCP scaffolds. This innovative approach not only bolstered the mechanical strength of TCP, but also allowed for the controlled modulation of its photothermal properties through adjustment of the LB content. This was demonstrated in both in vivo and in vitro experiments, where the scaffolds exhibited tumor inhibition capabilities. Furthermore, it was confirmed that NIR light promoted new bone formation, regardless of the presence or absence of LB.

TCP-related bifunctionality can also be attained through the binding of PTA to small molecules that promote osteogenesis. For instance, bone morphogenetic protein-2 (BMP-2) is known to stimulate the osteogenic differentiation of mesenchymal stem cells (MSCs) [183], Similarly, insulin-like growth factor I (IGF-1) is widely recognized as an osteogenic inducer due to its ability to enhance the osteogenic activity of bone morphogenetic protein-6 (BMP-6), particularly playing a pivotal role in bone healing within bone defects [184]. Liangjie Lu et al. [185] successfully fabricated an MPBI@β-TCP scaffold that involved the integration of bone growth factors BMP-2 and IGF-1 into a β-TCP scaffold coated with MoS_2_ using the curing agent polydopamine (PDA). MoS_2_ served as the PTA to induce a photothermal effect. Through in vitro co-cultures with MSCs and an established mouse femoral defect model experiment, it was observed that the osteogenic potential of MSCs was significantly stronger compared to other control groups. Moreover, the MPBI@β-TCP scaffold exhibited favorable anti-inflammatory and pro-angiogenic effects. Simultaneously, in a tumor-bearing mouse model, the tumor cell eradication rate reached an impressive 80.68% when exposed to near-infrared light (808 nm, 1 w/cm^−2^, 10 min) irradiation.

TCP-related bifunctional scaffolds can also serve as drug carriers for precision drug therapy. In 2020, Hongshi Ma et al. [186] successfully developed Cu-containing mesoporous silica nanosphere-modified β-tricalcium phosphate (Cu-MSN-TCP) scaffolds as copper (Cu) belongs to the category of PTAs and can induce a photothermal effect. Furthermore, Cu ions can promote angiogenesis and cooperate with β-tricalcium phosphate (TCP) in fostering bone regeneration. Mesoporous silica nanospheres (MSNs) serve as carriers for drugs and growth factors. Although the authors did not specifically investigate drug delivery using MSNs, these carriers were effectively integrated into the bifunctional scaffolds. In 2021, Dang, W. et al. [187] proposed an alternative approach. They utilized poly(D, L-lactide) (PDLLA) as a medium and combined DOX (doxorubicin) and TN (tanshinone) within the scaffold through a soaking and drying method. TN not only synergizes with TCP to promote bone regeneration, but also acts as a PTA. Moreover, by adjusting the drug concentration during immersion, the DOX content within the scaffold can be flexibly controlled, enabling precise PTT and localized controlled-release chemotherapy. This approach has demonstrated a pronounced synergistic effect in tumor cell destruction.

### 5.2. Nano-Hydroxyapatite (n-HA)

Hydroxyapatite (HA), with the chemical formula Ca_10_(PO_4_)_6_(OH)_2_, serves as the predominant inorganic component of human bone. Since its initial use as a graft material in 1971, HA has garnered significant attention in research due to its commendable biocompatibility [188]. Notably, HA implants can directly integrate with bone at the junction interface without forming fibrous interfaces. HA exhibits osteoinductivity and has no immunogenicity. However, pure HA has limitations including restricted load-bearing capacity, prolonged degradation time, high fragility, and a limited ability to recruit cells and stimulate angiogenesis [189,190]. To address these limitations, researchers have explored the potential of nano-hydroxyapatite (n-HA) as an improvement over pure HA. Compared to pure HA, n-HA exhibits enhanced mechanical properties, improved material density, and increased fracture toughness [191]. It also has superior biocompatibility, promotes osteogenesis more effectively, and can inhibit tumor cell growth [192].

Currently, none of the implants utilized in clinical practice possess tumor prevention and treatment capabilities. Consequently, biomaterials are frequently engineered as carriers for tumor treatment drugs. One notable advantage is the potential for surface modification of the carrier material with targeting groups, enabling precise drug delivery to the tumor site. When n-HA is employed as a carrier for PTAs, it introduces a novel, targeted, and noninvasive approach to inhibit tumor cells while concurrently promoting new bone formation.

Hence, the combination of n-HA and PTA has a dual function. Transition metal carbides and nitrides (MXenes) are a novel class of 2D inorganic nanomaterials that possess ultrafine structures and versatile functionalities. Guannan Zhang et al. [193] developed a three-dimensional porous n-HA/g-C3N4/MXene scaffold and showed that n-HA could further enhance the synergistic antitumor function of photodynamic and photothermal therapies. Additionally, the scaffolds were capable of eradicating osteosarcoma cells in just 10 min at a mild temperature of 45 °C. In addition, this multifunctional scaffold enhanced osteogenic activity with NIR. A 3D-printed PEEK/graphene nanocomposite scaffold was coated with a hydroxyapatite layer containing antibiotics and/or anticancer drugs. The graphene nanosheets within the scaffold acted as efficient PTAs. In vitro, the bioactive hydroxyapatite coating substantially increased stem cell proliferation. In vivo, it promoted the formation of new bone tissue. Furthermore, the inclusion of antibiotics and anticancer drugs enabled the elimination of drug-resistant bacteria and the destruction of osteosarcoma cancer cells. Notably, the treatment’s efficacy could be further augmented through on-demand laser-induced heating [194].

Chitosan (CS), a natural polycationic linear polysaccharide derived from lignin deacetylation, has been applied in various tissue engineering biomaterials due to its numerous advantages including biocompatibility, biodegradability, antibacterial properties, hemostasis, and adhesive qualities [195,196,197]. However, when used in scaffolds by itself, CS exhibits poor mechanical strength and undergoes rapid degradation [198]. To address these limitations, CS is often combined with nano-hydroxyapatite (n-HA) to create CS/n-HA composites, which have been extensively researched. Compared to CS alone, CS/n-HA composites offer improved mechanical properties, maintain favorable porosity and biocompatibility, enhance the differentiation and mineralization of mesenchymal stem cells by upregulating osteogenic genes, and demonstrate superior bone regeneration potential in vivo. Additionally, they have a beneficial synergistic effect with antibacterial properties [196,199]. However, CS/n-HA lacks a good inhibitory effect on tumor growth, and it could be improved by combining CS/n-HA with PTAs to construct multifunctional composites.

Liang Ma et al. [200] loaded GO and n-HA onto CS to create an n-HA/GO/CS scaffold. Their findings confirmed that this scaffold not only exhibited tumor-inhibiting and bone-promoting effects, but also demonstrated superior wound healing and hemostatic properties when exposed to NIR light. These effects were verified through a Kunming mouse skin defect model and blood coagulation experiments. The n-HA/GO/CS scaffold capitalized on the advantages of each component and addressed the limitations of individual components. HA/polydopamine (PDA)/carboxymethyl chitosan (CMCS) yields similar results, as each component synergistically enhances bone promotion and inhibits tumor growth. PDA, an organic PTA, is particularly noteworthy for its high photothermal efficiency [201]. CMCS, a chitosan derivative with improved water solubility, further enhances the scaffold’s properties. Mengyu Yao et al. [202] constructed a HA/PDA/CMCS scaffold, confirming its superior photothermal efficacy in bone tumor inhibition and enhanced osteogenic capabilities compared to a control group lacking PDA or NIR exposure. A mechanical property analysis and scaffold morphology assessment revealed that PDA significantly improved the rheological properties of the slurry, and increased the mechanical strength, surface potential, and water absorption properties of the composite scaffold.

Nanotechnology offers a promising solution in which to address infection-related issues in implanted scaffolds, and incorporating photothermal properties can further enhance their antibacterial capabilities [199,203]. In 2018, Lu et al. [204] introduced zero-dimensional CDs into a chitosan/nano-hydroxyapatite (CS/n-HA) scaffold, forming CS/n-HA/CD. CD synthesis is cost-effective and associated with relatively low biological toxicity, making it a valuable promoter of mesenchymal stem cell osteogenesis. However, its application in three-dimensional bone scaffolds has been limited [205]. Notably, CS/n-HA scaffolds can convert light energy into heat energy, effectively enabling PTT. These scaffolds have exhibited excellent photothermal efficiency, with the central temperature reaching up to 51.4 °C, even under NIR light exposure in tumor-bearing mice. Furthermore, CD-doped scaffolds can promote the adhesion of rat bone marrow-derived mesenchymal stem cells (rBMSCs) through focal adhesion and actin cytoskeleton pathways. In antibacterial experiments, this scaffold has demonstrated robust antibacterial properties under NIR irradiation, aligning with results from Choon Peng Teng et al., who utilized NIR-absorbing gold nanocrosses for the effective photothermal ablation of multidrug-resistant bacteria [206]. The underlying mechanism may involve the high temperature eliminating a portion of the bacterial biofilm.

## 6. Conclusions and Outlook

This article has provided a review of recent research progress in the application of PTT for the treatment of bone tumors. While PTT has been explored in various ways and has yielded promising experimental results, the majority of PTT research has centered on the design and development of PTAs. However, the translation of PTT into clinical applications is still a work in progress, with several shortcomings that require improvement. Furthermore, common challenges persist across various application methods, necessitating solutions.

The first issue pertains to light penetration in PTT. Currently, most PTA applications require a wavelength of 808 nm, which falls within the NIR-I window (NIR-I, 650–950 nm). However, this window has inherent limitations including limited laser light penetration (approximately 1–2 cm) and a maximum permissible exposure of 0.33 W cm^−2^. In contrast, the NIR-II window (1000–1700 nm) is more attractive for biomedical applications due to its capacity for deeper tissue penetration (beyond 2 cm) and significantly higher maximum permissible exposure (1 W cm^−2^) [207,208,209]. In osteosarcoma treatment, NIR-II PTAs have been reported such as Wesselsite (SrCuSi_4_O_10_) nanosheets (SC NSs), FePSe3, and titanium dioxide nanorods (TiO_2_ NRs) [179,210,211]. These materials can be linked with active small molecules like hyaluronic acid (HA) and bone morphogenetic protein-2 (BMP-2). When exposed to near-infrared II light, they enable the extensive hyperthermia-induced ablation of deep-seated osteosarcoma and enhance vascularized bone regeneration in vivo. This exploration of NIR-II excitation wavelengths for PTAs offers a promising solution to the challenge of tissue penetration of the light source.

The second challenge involves determining the optimal light timing and frequency in PTT treatment. In in vivo PTA experiments, near-infrared light irradiation has shown potential in inhibiting the growth of bone tumor cells. However, the questions of when to administer light, how frequently, and for what duration remain unresolved. Current reports often base these determinations on ideal conditions for specific PTA materials. The diversity of PTA materials and the variability of illumination conditions complicate their clinical translation and application.

Finally, concerns exist around the biological toxicity and long-term metabolic safety of PTT. Toxicity remains a primary concern with nanomaterials due to their minuscule size, which allows them to breach physiological barriers and pose potential health risks [212]. Nanomaterials can generate free radicals that damage cell membranes, organelles, and DNA [213]. Some PTA materials, like certain carbon-based substances, are inherently toxic. Surface modification for reducing toxicity is a contemporary solution, but the long-term metabolic fate of these materials cannot be overlooked.

PTT-based targeted therapy: Presently, many nanomaterial targeting mechanisms rely on the EPR effect. However, the clinical relevance and effectiveness of EPR-based targeting remain inconclusive, necessitating further investigation [214]. Determining how to achieve precise drug delivery remains the central focus of targeted therapy.

PTT-based immunotherapy: The immune mechanisms of PTT in bone tumors remain unclear, and there is limited research on combining PTT with immunotherapy in the context of bone tumors. Furthermore, immunotherapy is costly, and patient responses vary significantly. Investigating how the high-temperature environment post-light exposure affects immune cells and the tumor microenvironment presents a major research challenge.

Finally, all of the studies discussed in this review are at the preclinical stage. We must continue to explore the potential of synergistic therapies through a multidisciplinary, multipathway approach. It is anticipated that future clinical investigations in PTT will yield results that open new possibilities for personalized cancer treatment.

## Figures and Tables

**Figure 1 ijms-25-04139-f001:**
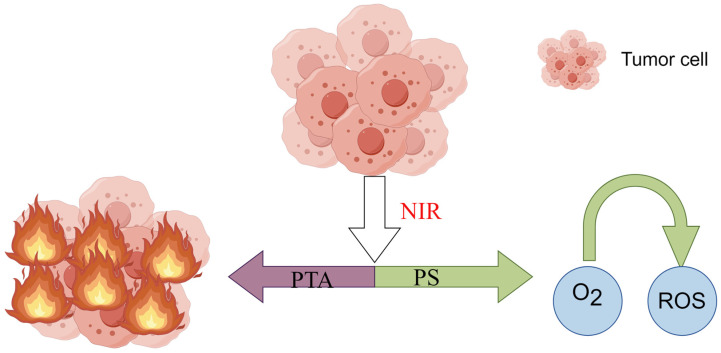
Mechanisms of PTT and PDT. PTT: relies on the efficient conversion of light energy into heat energy by the PTA to exert cytocidal effects on tumor cells. PDT: activates PS to generate ROS, leading to apoptosis, necrosis, or autophagy, ultimately resulting in cytotoxicity and cell death. Created with Figdraw (SWSWO1b012). Before PTA was extensively researched, cancer was often treated separately with laser equipment through methods such as laser interstitial thermal therapy by directly inserting laser fibers into the tumor and using PTT to directly kill the tumor. This method is used for prostate cancer, liver tumors [13,14], and is also known as laser ablation. However, like other ablation methods, it cannot accurately irradiate malignant tumors and protect normal tissues, and direct laser ablation requires high power, so there are still significant limitations in using light-induced heat directly. In 2003, Hirsch et al. [15] developed a metallic PTA in which silica nanoparticles were enveloped by small gold colloids to create gold–silica nanoshells, which were subsequently modified with polyethylene glycol (PEG) to maintain the stability of the nanoshell colloid. After exposure to NIR light (820 nm, 35 W/cm^2^), human breast carcinoma cells cultured with the obtained PTA lost viability, while cells cultured with only NIR light or PTA viability. The study suggests that the low-power near-infrared light irradiation of PTA can noninvasively kill tumor cells, and the surrounding normal healthy tissues are not affected. PTT is a noninvasive, controllable, and targeted strategy by which to eliminate tumor cells; therefore, it has been widely studied for bone cancer therapy in the past decade [16].

**Figure 2 ijms-25-04139-f002:**
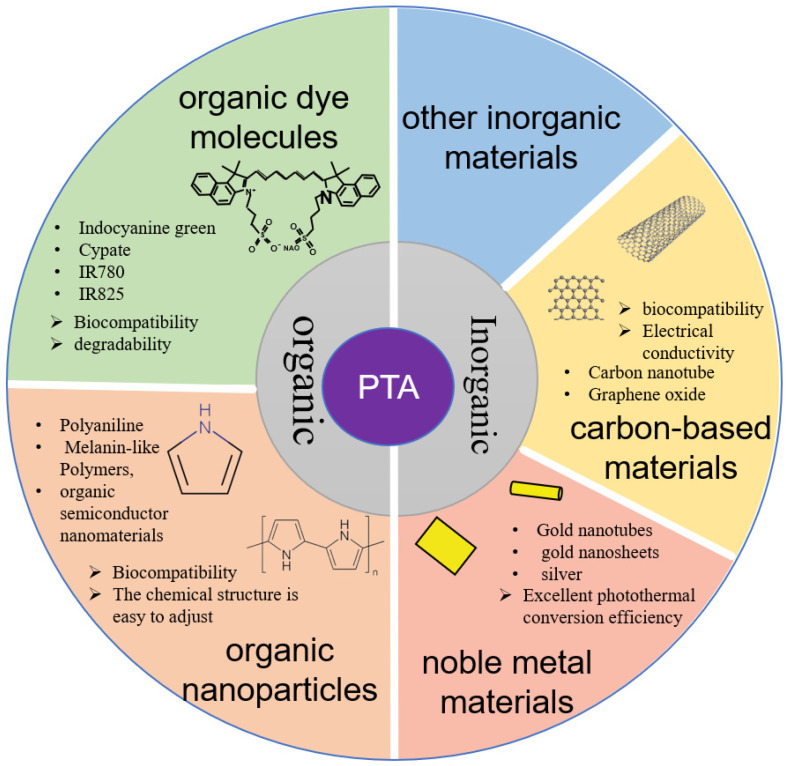
Classification and advantages of PTAs.

**Figure 3 ijms-25-04139-f003:**
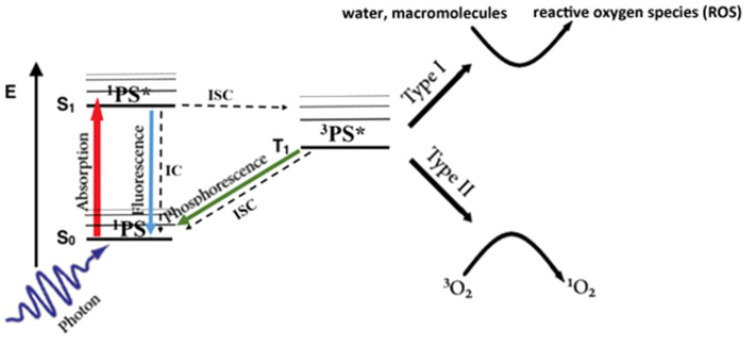
Mechanisms of PDT. ^1^PS (S0): the ground-state PS; 1PS* (S1): electronic singlet state; 3PS* (T1): a triplet state; Type I: 3PS* (T1) reacts with biomolecules to form free radical species; Type II: ground-state molecular oxygen, ^3^O_2_, to form highly-reactive singlet oxygen, ^1^O_2_. Radiative transitions are shown as colored arrows and nonradiative transitions as black/dashed arrows. Reproduced with permission from Reference [24], Copyright 2023, *Front. Bioeng. Biotechnol.*

**Figure 4 ijms-25-04139-f004:**
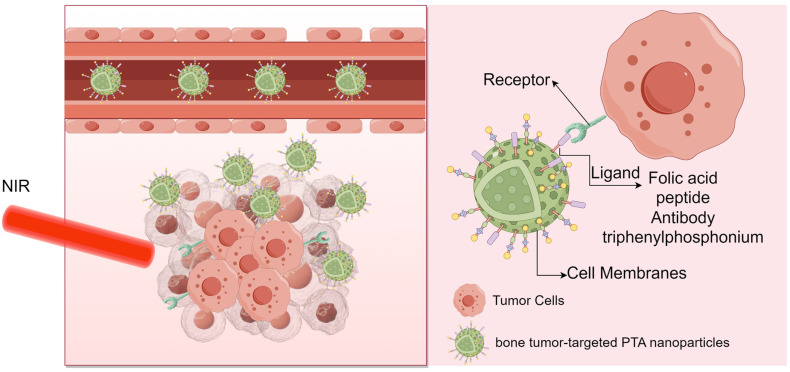
Target antitumor treatment with PTT. Created with Figdraw (USUYA04846).

**Figure 5 ijms-25-04139-f005:**
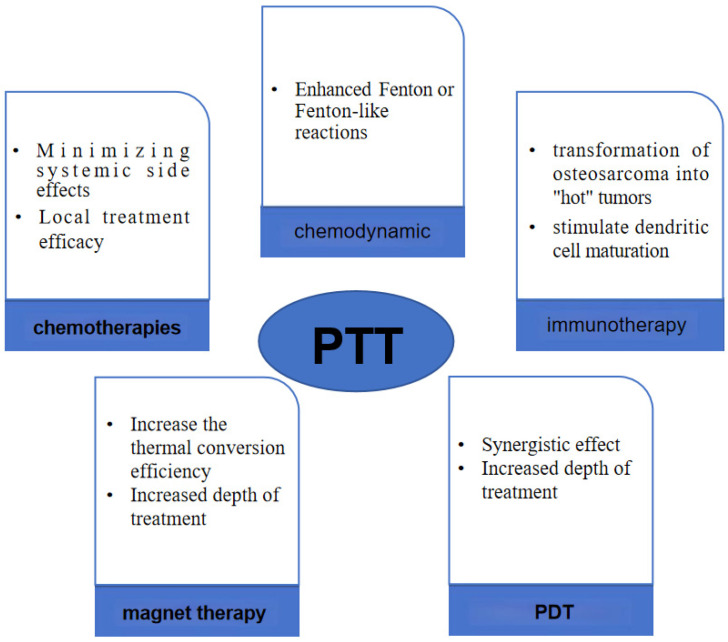
PTT combined multiple therapies.

**Figure 6 ijms-25-04139-f006:**
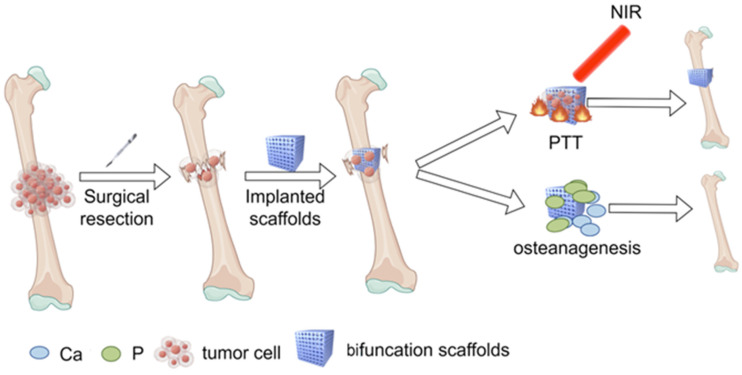
The bifunctional scaffold inhibits tumor growth and promotes osteogenesis. Created with Figdraw (WSYOS98b7b).

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
