# Peer review of "Advancements in Photothermal Therapy Using Near-Infrared Light for Bone Tumors"

_ijms, 2024, doi:10.3390/ijms25084139_

Round 1
Reviewer 1 Report
Comments and Suggestions for Authors
The following comments needs to be addressed before accepting for the final publication.
1. There are some formatting errors in the manuscript like the abbreviation of some items is repeating. For example, PTA was first abbreviated in Line 78, but again repeating in Line 93. Likewise the same in Figure 1 caption also repeating. PDT in like 78 and Line 111. Also, PS is repeating in Figure 1 caption, and from Lines 111 to 135. Please check all over the manuscript.
2. As similar to PTT and PDT, there is some other kind of non-invasive therapy called the magnetically controlled therapy or magnetic therapy. There are some researchers who applied both magnetic therapy in combination with PDT, can the authors include some sections related to both these therapies towards the controlling or therapy of bone tumors. It will be more beneficial to the readers as they both are non-invasive and sustainable.
3. The mechanism associated with the PDT, i.e. the movement of electrons in the valence state to the excited states mentioned in Lines 103-109, it will be good if the authors can make a schematic diagram for this phenomenon. There are several articles published on this mechanism.
4. Figure 2 is very poor in terms of both information that it contains and also the visibility. The authors need to extend this figure with much more information like each kind of material is applicable/specific to what kind of cancer, advantages etc along with its quality.
5. It will be good if the authors can incorporate a separate section mentioning the ideal characteristics (physical properties) that an active photosensitizer and the NIR/laser/light frequency/wavelength range to be incorporated in order to generate PTT with very high efficacy to destruct the cancer diagnosed cells.
6. It will be useful if the authors can include a section containing the biological effects that can bring to both cancer and non-cancer cells following PTT effects. As it can enable to design no ways to overcome the limitation and improve the efficacy of the PTT method.
Comments on the Quality of English LanguageEnglish proof reading is required. There are several grammatical errors and the abbreviating some terms again and again.
Reviewer 2 Report
Comments and Suggestions for Authors The authors submitted a manuscript that summarizes the achievements in photothermal therapy using near-infrared light for bone tumors. This manuscript requires major changes. Firstly, not only linguistic correction of this manuscript is required, but also the editorial side (spaces, superscripts and subscripts). Secondly, the figures shown are of poor quality and their correction is required. It is necessary to adapt these figures to the editorial requirements of the journal. Thirdly, it is proposed to familiarize yourself with the latest data relating to the use of curcumin for the destruction of pathogenic cells.Comments on the Quality of English Language
This manuscript requires major changes